# SULF1 suppresses Wnt3A-driven growth of bone metastatic prostate cancer in perlecan-modified 3D cancer-stroma-macrophage triculture models

**Fabio Henrique Brasil da Costa**[1,2], **Michael S. Lewis**[3], **Anna Truong**[4], **Daniel D. Carson**[1,5], **Mary C. Farach-Carson**[1,2,6¤] *

**1** Biosciences Department, Rice University, Houston, TX, United States of America, **2** Department of Diagnostic and Biomedical Sciences, The University of Texas Health Science Center School of Dentistry, Houston, TX, United States of America, **3** Department of Pathology and Medicine, Cedars-Sinai Medical Center, West Hollywood, CA, United States of America, **4** Department of Chemistry, Rice University, Houston, TX, United States of America, **5** Department of Genetics, The University of Texas MD Anderson Cancer Center, Houston, TX, United States of America, **6** Department of Bioengineering, Rice University, Houston, TX, United States of America

¤ Current address: Department of Diagnostic and Biomedical Sciences, The University of Texas Health Science Center School of Dentistry, Houston, TX, United States of America

* Mary.C.FarachCarson@uth.tmc.edu

**Data Availability Statement:** All relevant data are within the manuscript and its Supporting Information files.

## Abstract

Bone marrow stroma influences metastatic prostate cancer (PCa) progression, latency, and recurrence. At sites of PCa bone metastasis, cancer-associated fibroblasts and tumor-associated macrophages interact to establish a perlecan-rich desmoplastic stroma. As a heparan sulfate proteoglycan, perlecan (*HSPG2*) stores and stabilizes growth factors, including heparin-binding Wnt3A, a positive regulator of PCa cell growth. Because PCa cells alone do not induce CAF production of perlecan in the desmoplastic stroma, we sought to discover the sources of perlecan and its growth factor-releasing modifiers SULF1, SULF2, and heparanase in PCa cells and xenografts, bone marrow fibroblasts, and macrophages. SULF1, produced primarily by bone marrow fibroblasts, was the main glycosaminoglycanase present, a finding validated with primary tissue specimens of PCa metastases with desmoplastic bone stroma. Expression of both *HSPG2* and *SULF1* was concentrated in αSMA-rich stroma near PCa tumor nests, where infiltrating pro-tumor TAMs also were present. To decipher SULF1's role in the reactive bone stroma, we created a bone marrow biomimetic hydrogel incorporating perlecan, PCa cells, macrophages, and fibroblastic bone marrow stromal cells. Finding that M2-like macrophages increased levels of *SULF1* and *HSPG2* produced by fibroblasts, we examined SULF1 function in Wnt3A-mediated PCa tumoroid growth in tricultures. Comparing control or *SULF1* knockout fibroblastic cells, we showed that SULF1 reduces Wnt3A-driven growth, cellularity, and cluster number of PCa cells in our 3D model. We conclude that SULF1 can suppress Wnt3A-driven growth signals in the desmoplastic stroma of PCa bone metastases, and *SULF1* loss favors PCa progression, even in the presence of pro-tumorigenic TAMs.

**Funding:** This work was funded by a grant from the National Institutes of Health/National Cancer Institute, P01CA098912. Mr. Brasil da Costa was supported in part by the Brazilian Coordination for the Improvement of Higher Education Personnel (CAPES). The funders had no role in study design, data collection and analysis, decision to publish, or preparation of the manuscript.

**Competing interests:** The authors have declared that no competing interests exist.

# Introduction

Prostate cancer (PCa) is the most common and second leading cause of cancer-related deaths among men [1]. PCa demonstrates metastatic tropism for bone marrow; over 80% of PCa patients who succumb to disease harbor bone metastases at autopsy [2]. At the metastatic stage, PCa often develops androgen insensitivity and becomes treatment-resistant [3]. For many years, the involvement of host cells during cancer progression was neglected. Now, multiple lines of evidence support a role for stromal and immune cells in the transition of PCa from indolent to lethal phenotype [4, 5]. A dynamic cross-talk between PCa cells, cancer-associated fibroblasts (CAFs), and tumor-associated macrophages (TAMs) impacts cancer behavior and disrupts normal tissue homeostasis [6–9]. Features of desmoplasia include increased deposition of extracellular matrix (ECM) components, growth factors, and ECM-remodeling enzymes, as well as recruitment of immune cells, including macrophages [10, 11]. The landscape of macrophage infiltration and phenotype in bone metastases of PCa has been explored in murine disease models [12] and tissue specimens from warm biopsy specimens [13], but never in a human biomimetic system where macrophages can directly interact with PCa and fibroblastic stromal cells simultaneously.

Matrix remodeling in the tumor reactive stroma classically involves elevated deposition of the heparan sulfate proteoglycan 2 (*HSPG2*), perlecan [14, 15]. The N-terminal domain of perlecan contains attachment sites for the glycosaminoglycan (GAG), heparan sulfate [16, 17] that serves as a high capacity depot to store and stabilize heparin-binding growth factors (HBGFs), including Wnts [18–20]. The HBGF Wnt3A mediates PCa tumorigenesis and proliferation [21], and was implicated in pathways that favor PCa bone metastasis [22]. Wnt3a is stored and stabilized by 6-O-sulfate-bearing heparan sulfate chains [23] attached to proteoglycans, such as perlecan, in the matrix. The signaling of Wnt3a at the cancer cell surface is facilitated by heparan sulfate on other proteoglycans, such as syndecan or glypican, acting as signaling co-receptors. At the present, it is not well understood how heparan sulfate and its enzymic modifiers, including those that act on 6-O-sulfate, control Wnt3A bioavailability and receptor activation in the context of PCa progression in bone.

Two mechanisms have been described for HBGF release by the heparan sulfate-modifying enzymes heparanase (HPSE), and sulfatases 1 and 2 (SULF1 and SULF2) [24, 25]. Catalytically, HPSE is an endo-$\beta$-glucuronidase that cleaves heparan sulfate chains yielding relatively large fragments (5–10 kDa or 10–20 sugar units) [26] that stabilize growth factor binding to their receptors in a functional ternary complex [27]. SULF1 and SULF2 are closely related 6-O endosulfatases [24, 28], that are secreted or stay peripherally associated with proteoglycans at the cell membrane [29]. Enzymatic removal of 6-O-sulfate from heparan sulfate by SULFs in the ECM releases bound growth factors from HSPGs, which may activate or suppress signaling, depending on the context [30–32]. In the cell surface glycocalyx, removal of 6-O-sulfate diminishes the co-receptor functions of heparan sulfate proteoglycans and blocks growth factor signaling. Hence the SULFs have been considered as locally acting tumor suppressors in many carcinomas [33]. The opposing mechanisms of action by HPSE and SULFs make it challenging to predict signaling outcomes, but context plays a crucial role in how cells in tissue will respond. In PCa, the function and expression of heparan sulfate modifiers that can impact signaling by Wnt3A remain unstudied.

To explore the control of Wnt3A signaling in a physiological mimic of the bone metastatic environment, we developed a perlecan-modified hydrogel triculture system that supports the growth and interactions among PCa cells, bone marrow stromal fibroblasts, and macrophages. We obtained rare patient specimens of bone metastatic PCa that allowed us to validate our findings in the hydrogel model. Furthermore, while we previously reported that the perlecan

core protein accumulates in the reactive stroma of primary prostate tumors [15], its cellular source(s) remained undetermined. In this study, we identified the cells that produce perlecan and its major enzyme modifier SULF1 and their respective roles in regulating Wnt3A-induced growth of metastatic PCa cells growing in perlecan-rich desmoplastic stroma, such as would occur at sites of bone metastasis.

## Results

### *SULF1* is primarily expressed by bone marrow fibroblasts *in vitro*

Basal transcript levels of *SULF1*, *SULF2* and *HPSE* were measured among a series of cell types of interest in the context of PCa bone metastases. These included PCa cell lines (LNCaP, C4-2, C4-2B, PC-3), patient-derived xenografts (MDA PCa 118b and 183 PDXs), bone marrow fibroblasts (HS27A and HS-5), primary human bone marrow stromal cells (BMSCs), prostate stromal fibroblasts (WPMY-1), and primary macrophages (either unpolarized (M0-M$\phi$) or polarized towards a pro-tumor phenotype (M2-M$\phi$)). We found that the mRNA levels of *SULF1* were 150 to 200 times higher in HS27As and BMSCs compared to PCa cells and macrophages (Fig 1A), while *SULF2* mRNA levels were both substantially lower and similar across all cell types tested (Fig 1B). *HPSE* mRNA levels were also generally comparable among all cells, except for HS27A cells, which displayed considerably higher levels (Fig 1C). Nevertheless, when comparing the levels of each mRNA in bone marrow fibroblasts, we noted that *SULF1* mRNA levels were approximately 12-fold and 50-fold higher than those of *HPSE* and *SULF2* in HS27A cells, respectively; and nearly 60 fold and 30 fold higher than *HPSE* and *SULF2* in BMSCs, respectively (S1 Fig). We made multiple attempts to assess levels and distribution of SULF1 protein both by immunostaining and western blotting. We tested all commercially available antibodies for SULF1 that were generated with unique immunogens. Regardless of the antibody used, SULF1 antibody assays provided results inconsistent with SULF1's size, known cellular expression assessed by PCR or resulted in multiple non-specific bands (S2 Fig). Most importantly, no antibodies accurately reported the loss of *SULF1* expected in *SULF1*-KD or KO cells validated by loss of transcripts in PCR assays. Therefore, based on mRNA expression, the data indicate that bone marrow fibroblasts are significant sources of SULF1 and that SULF1 is a major heparan sulfate-modifying enzyme in this microenvironment.

### *SULF1* and *HSPG2* are mainly produced in the desmoplastic tumor stroma

Next, we investigated transcript levels of *SULF1* and *HSPG2* in human PCa bone metastases to determine their cellular sources *in vivo*. Multiplexing of chromogenic RNA in situ hybridization (RISH) and immunofluorescence on the same sections, in series, allowed the co-staining of mRNAs with the activated fibroblast and epithelial markers alpha-smooth muscle actin ($\alpha$SMA) and E-cadherin (E-cad), respectively. The levels of *SULF1* (Fig 2A1–2A3) and *HSPG2* (Fig 2B1–2B3) transcripts were highest in the cells in the stroma between cancer nests. Because transcripts and the proteins that they encode are concentrated in different cellular compartments, a direct colocalization measurement is not possible. Instead, we quantified the transcript levels we detected in each cell population using Imaris software, as described in Materials and Methods. Quantification of the *SULF1* mRNA shows that approximately 95% of the signal is confined to $\alpha$SMA$^+$ activated fibroblasts, and is nearly undetected in E-cad$^+$ tumor cells (Fig 2C). Also, while *HSPG2* mRNA was mostly expressed by CAFs, close to 20% of E-cad$^+$ tumor cells were *HSPG2*$^+$ (Fig 2D). This pattern was detected in all other patient samples, and additional representative images and controls can be found in the supplementary information (S3–S6 Figs).

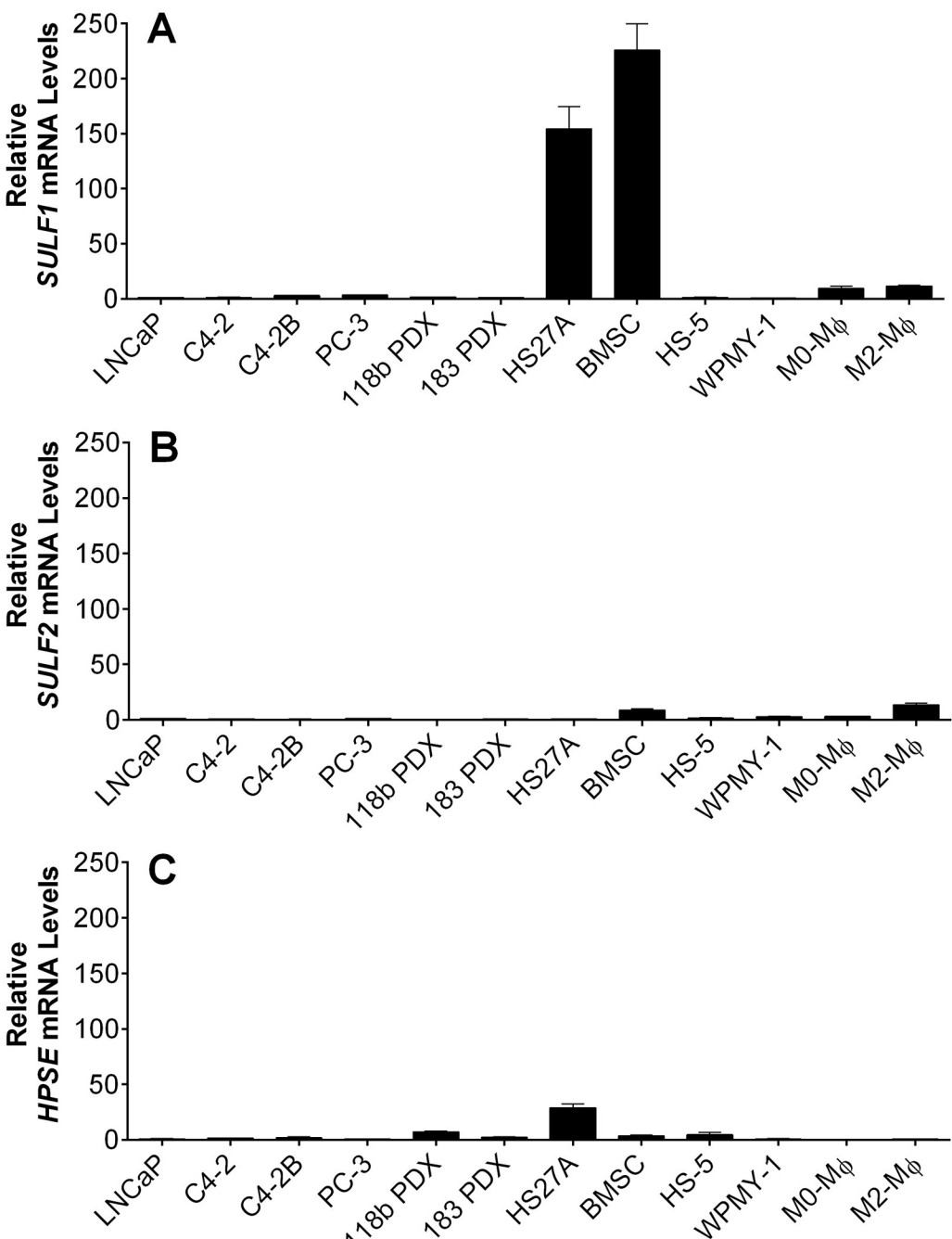

**Fig 1. Basal mRNA levels of *SULF1*,*SULF2*, and *HPSE* a variety of cancer, stromal and immune cells.** PCa cells (LNCaP, C4-2, C4-2B, PC-3, 118B PDX and 183 PDX), stromal cells (HS27A, BMSC, HS5, WPMY-1), and unpolarized (M0-Mφs) or alternatively activated macrophages (M2-Mφs). Cells were prepared for RNA isolation and qPCR, as described in Materials and Methods. The expression of *SULF1* (A), *SULF2* (B) and *HPSE* (C) was normalized to that of *GAPDH*. Values obtained for LNCaP cells were arbitrarily set to 1 for comparison. Data shown represent the mean ±SD of at least three independent experiments.

Taken together, these data suggest that a robust infiltration of TAMs exists in bone metastasis of PCa, which show features of alternative activation—a phenotype that was replicated in our biomimetic hydrogels.

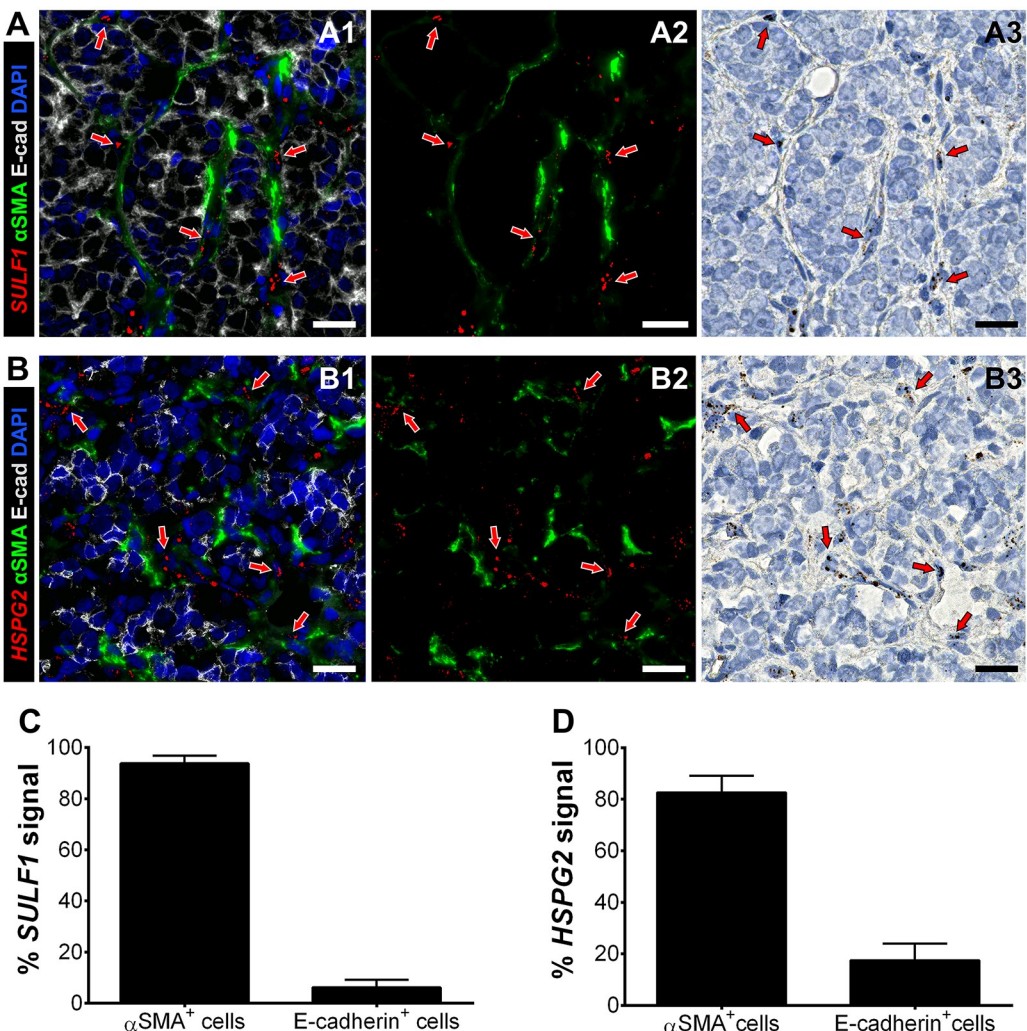

**Fig 2. Multiplexing of RISH-immunofluorescence for detection of *SULF1* or *HSPG2*, in tandem with stromal and epithelial markers.** *SULF1* (A1-3) and *HSPG2* (B1-3) expression was mainly detected in $\alpha$SMA+ CAFs (green) surrounding E-cad+ tumor clusters (gray). A3 and B3 show the real chromogenic mRNA signals, which were deconvoluted for quantification, and merged with the green channel for easier visualization (A2 and B2), as described in Materials and Methods. The percentage of SULF1+ (C) and HSPG2+ (D) was measured in the stromal ($\alpha$SMA+) and tumor (E-cad+) cell populations using Imaris, as described in Materials and Methods. A and B represent approximately 2.5X magnified regions of images acquired using a 40X objective, which are available as a whole in S2 Fig. Nuclei were counter-stained with DAPI (blue) and hematoxylin. Red arrows help indicate regions were the mRNA signal was detected. Scale bars represent 20 microns.

## Macrophages exhibit alternative activation in bone marrow metastases and in bone marrow-mimicking hydrogels

Immune cell recruitment is another critical element of the reactive stroma response. We investigated macrophage infiltration and phenotype in PCa bone marrow metastases for comparison with our biomimetic hydrogels. TAM infiltration was observed in all patients, displaying an alternatively activated phenotype (M2-M$\phi$), as indicated by CD163 and CD206 expression (Fig 3A and S6 Fig). Additionally, TAMs did not seem to form direct cell contacts with the cancer cells, but rather were dispersed in the stroma. Subsequently, we mimicked the

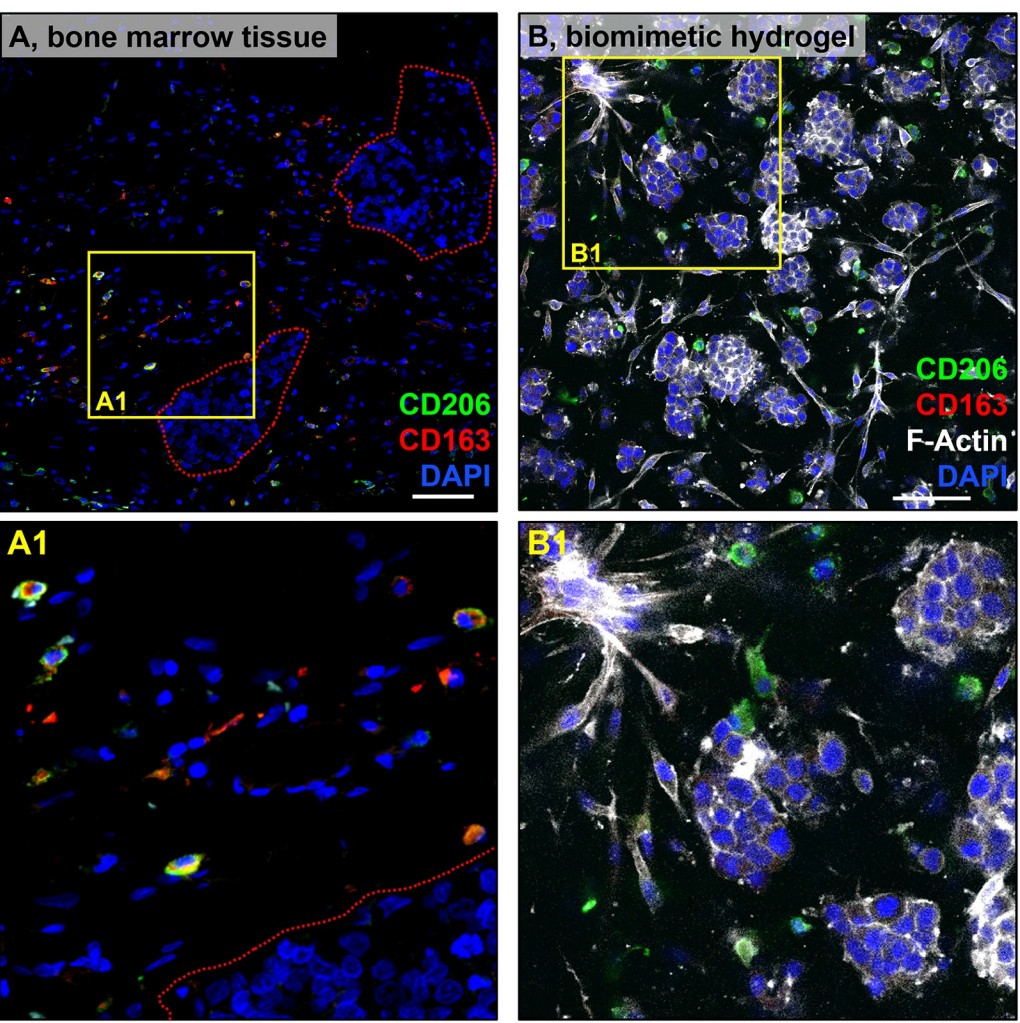

**Fig 3. Macrophage phenotype *in vivo* and *in vitro*.** Immunostaining of CD206 (green) and CD163 (red), with DAPI (blue) counterstain, in bone marrow containing PCa metastasis (femur) (A). Macrophages can be observed in the vicinity of the tumor nests (Red-dotted lines). C4-2B, HS27A cells, and macrophages were co-encapsulated within collagen type I and hyaluronic acid hydrogels, as described in Materials and Methods. N = 3 hydrogels per independent experiment. At day 9, samples were immunostained for CD206 (green) and CD163 (red), and counterstained with F-actin (white) and DAPI (blue) (B). Scale bar = 100 microns. Yellow solid lines indicate insets for each A and B, magnified in A1 and B1. Signal intensity was adjusted relative to negative controls in which the primary antibody was omitted. Data shown are representative of three independent experiments for each type of sample.

metastatic bone microenvironment using hyaluronic acid-collagen type I hydrogels to determine whether macrophages would polarize similarly as *in vivo*. In 3D tricultures of macrophages, C4-B, and HS27A cells, we observed that macrophages also expressed CD206, resembling TAMs *in vivo* (Fig 3B). CD206 expression was detected as early as 3 days after encapsulation (not shown). To demonstrate that the macrophages also were not polarizing towards an M1 phenotype, we also examined CD80 expression, which was not detected (S7 Fig). Taken together, these data show TAM infiltration in all PCa bone metastasis patient samples tested, and TAMs exhibited an M2-like phenotype, as indicated by CD163 and CD206 expression, which was partly replicated in our biomimetic hydrogels.

## *SULF1* and *HSPG2* transcript levels are significantly higher in BMSCs after exposure to M2-M$\phi$ conditioned medium

We explored whether M2-M$\phi$s had an impact on *SULF1* expression. Exposure of either HS27A or BMSCs to conditioned medium (CM) from M2-M$\phi$ for 48 hours led to a significant increase in *SULF1* mRNA levels, most notably a nearly 3-fold increase in primary BMSCs (Fig 4A). In contrast, levels of *SULF2* and *HPSE* mRNA remained unaltered, and exposure of C4-2B cells to conditioned medium from M2-M$\phi$ did not alter *SULF1* expression (S8 Fig). Likewise, steady-state mRNA levels of *HSPG2* were increased 3- and 5-fold in HS27A cells and primary BMSCs, respectively. Collectively, these data indicated that factors secreted by M2-M$\phi$s stimulate both *SULF1* and *HSPG2* expression by bone marrow fibroblasts.

## Loss of stromal *SULF1* significantly increases total PCa cellularity and cluster sizes in response to Wnt3a treatment

We used the biomimetic triculture model to assess the role of SULF1 in modulating PCa growth in response to Wnt3A, a key HBGF implicated in PCa progression. Given that we

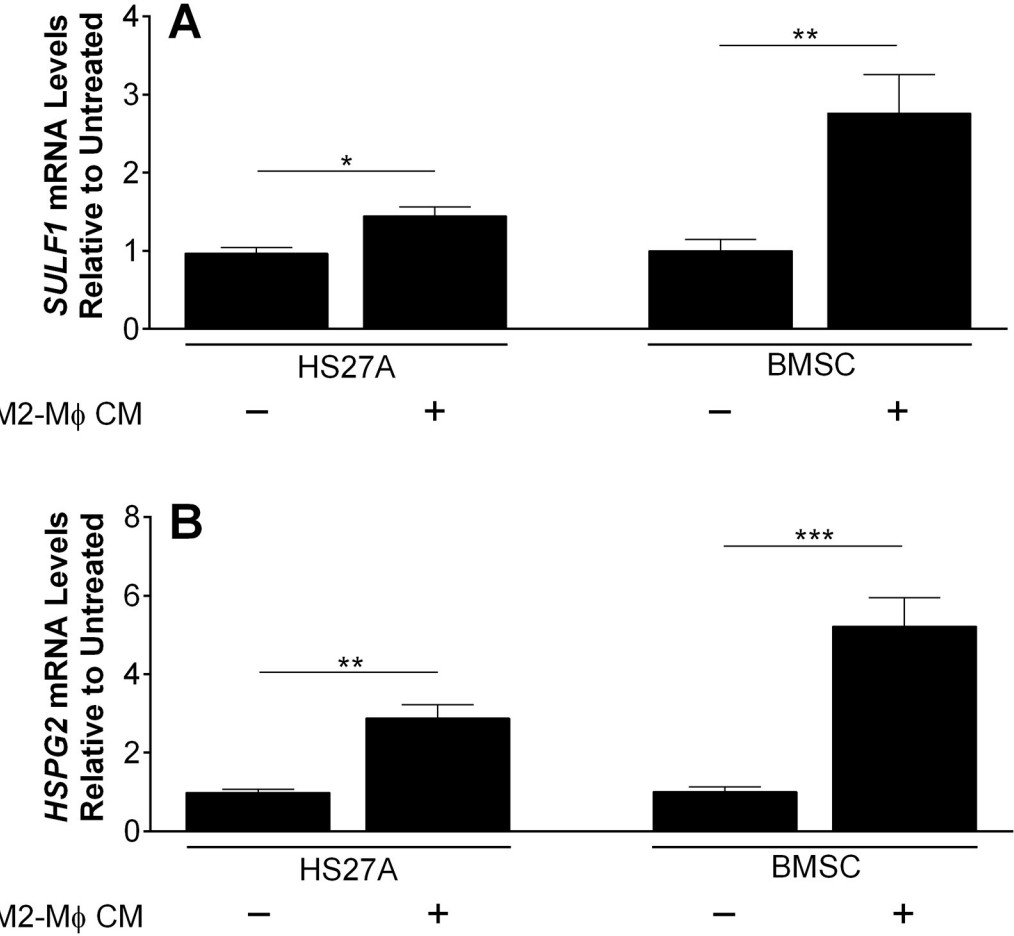

**Fig 4. Expression of (A) *SULF1* and (B) *HSPG2* mRNA by human bone marrow fibroblasts stimulated by CM of M2-M$\phi$s.** HS27A and primary bone marrow stromal cells (BMSC) were treated with fresh CM from M2-M$\phi$s for 48h and RNA was subsequently isolated for qPCR. The expression of *SULF1* and *HSPG2* mRNA was normalized to that of *GAPDH*. Values obtained for untreated cells (black columns) were arbitrarily set to 1 for comparison. Data shown represent the mean ±SD of three independent experiments. *, P < 0.05; **, P < 0.01; ***, P < 0.001.

found fibroblasts to be the primary source of SULF1, we used CRISPR-Cas gene editing to knockout (KO) *SULF1* expression in the HS27A bone stromal cell line, as described in Materials and Methods. We used hydrogels modified with heparan sulfate-bearing perlecan domain I (PlnDmI) to simulate perlecan's growth factor-stabilizing role in the ECM. Wnt3A was pre-conjugated onto PlnDmI before incorporation within the gel solution. This was intended to allow SULF1 to modify heparan sulfate on PlnDmI and regulate growth factor availability. In tricultures without Wnt3A, containing either wild-type-(WT) (Fig 5A1 and 5A2) or *SULF1*-KO-HS27A cells (Fig 5A5 and 5A6), PCa cells (E-cad⁺) displayed similar growth characteristics. However, in tricultures with *SULF1*-KO-HS27As and Wnt3A (Fig 5A7 and 5A8), there was a significant increase in C4-2B cellularity (Fig 5B) and tumoroid sizes (Fig 5C). Vimentin (Vim) staining was used to identify spatial distribution and morphology of HS27A fibroblasts (red) and macrophages (pseudo-colored in yellow, as described in Materials and Methods)—

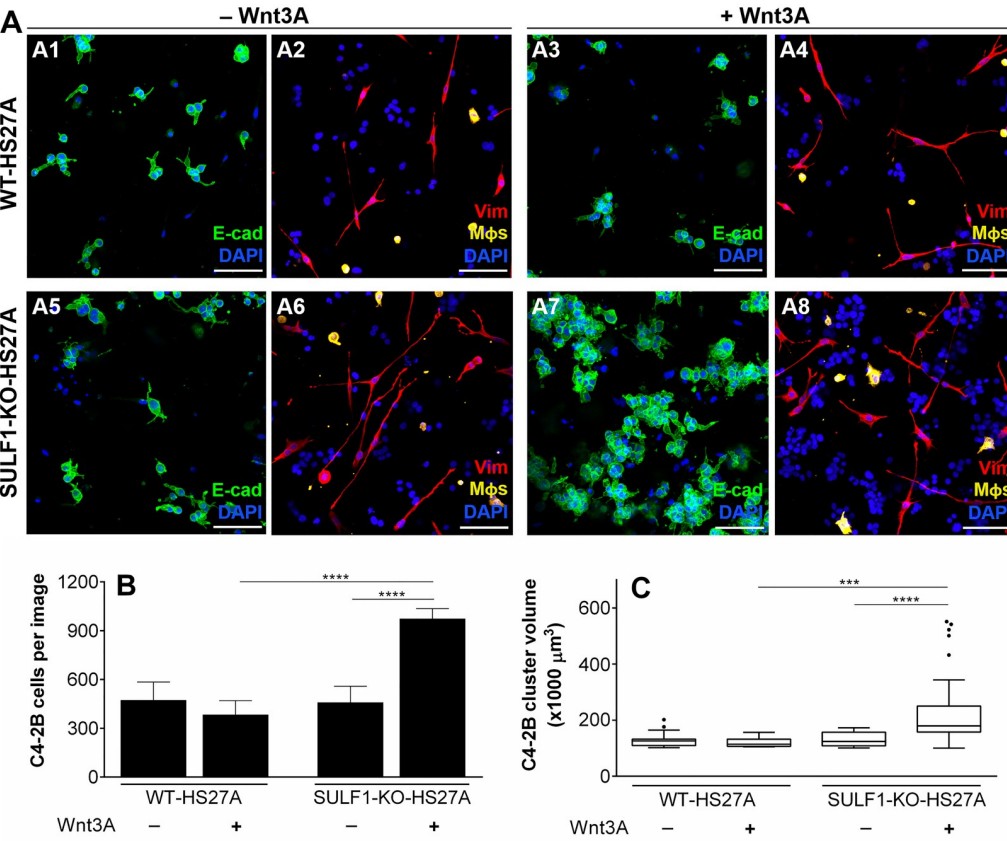

**Fig 5. Effect of Wnt3A treatment on tricultures of C4-2Bs, Wild-type (WT) or *SULF1* knockout (SULF1-KO) HS27As, and Mφs, encapsulated in biomimetic hydrogels.** Biomimetic collagen-HA hydrogels were prepared with the addition of human recombinant PlnDm1, pre-loaded with 200 ng of Wnt3A (+Wnt3A) or BSA (-Wnt3A), as detailed in Materials and Methods. In each hydrogel, 40.000 C4-2B, 10.000 HS27A and 5.000 Mφs were initially encapsulated. On day 5 after encapsulation, the hydrogels were fixed and prepared for immunostaining, as explained in Materials and Methods. Immunofluorescence stainings show E-cad (green) (A1, A3, A5 and A7), and vimentin (Vim; red) (A2, A4, A6, A8), with DAPI (blue) counterstain. E-cad and Vimentin signals were used to quantify total C4-2B cellularity (B) and cluster volume (C) *via* IMARIS software, as described in Materials and Methods. In A2, A4, A6 and A8, macrophages (vimentin-positive) were pseudo-colored in yellow according to cellular sphericity, as specified in Materials and Methods. Staining of CD206 also was performed in the hydrogel groups in this experiment (S10 Fig). Scale-bar = 100 microns. Data shown represent the mean ±SD of three independent experiments. In each experiment, N = 5 hydrogels for each of the four groups. ***, P < 0.001 and ****, P < 0.0001.

both cells did not seem to associate tightly with the PCa aggregates. No alterations in *SULF2* and *HPSE* mRNA levels were detected in *SULF1*-KO-HS27A cells (S9 Fig) compared to wild-type. Collectively, these data indicate that SULF1 is a negative regulator of the Wnt3A signaling pathway in our bone marrow biomimetic triculture system.

## Discussion

3D systems mimic mechanical properties and ECM cues of the *in vivo* tumor microenvironment, improving our ability to model many aspects of cancer cell behavior. We designed a custom hydrogel culture model to study both cellular and extracellular components in ways that recapitulate cellular interactions in the PCa bone-metastatic microenvironment. The makeup of the hydrogels included elements that are abundant in the bone marrow, such as collagen type I and hyaluronic acid. In addition, we incorporated recombinantly produced perlecan domain I (PlnDmI), which harbors up to three heparan sulfate chains, simulating the growth factor sequestration and delivery functions of the full molecule. In this system, recombinant Wnt3A was pre-bound to PlnDmI to explore the impact of SULF1 activity on PCa cell responses to this critical growth factor previously identified to play a vital role in PCa growth [22]. Additionally, our hydrogels were optimized to simultaneously support the growth and viability of a variety of cell types present in the reactive metastatic PCa bone microenvironment, allowing us to create direct tricultures by encapsulating bone marrow fibroblasts and macrophages together with PCa cells.

Fibroblasts react to local hyperplasia and inflammation by initiating a desmoplastic response [34, 35]. Increased deposition of perlecan is a major feature of this response in PCa [4, 15, 36]. Here, we provided complementary evidence that unambiguously shows CAFs are the primary sources of perlecan *in vivo*. We also previously demonstrated that *HSPG2* expression is up-regulated at the promoter level by inflammatory cytokines transforming growth factor $\beta$1 (TGF$\beta$1) and tumor necrosis factor $\alpha$ (TNF$\alpha$) [15]. While bone marrow fibroblasts cells make and produce TGF$\beta$1, neither they nor PCa cells produce enough TNF$\alpha$ to stimulate *HSPG2* expression in the desmoplastic stroma. Therefore, we hypothesized that a third cell type present in the stroma was needed to account for perlecan accumulation in the reactive stroma.

In many carcinomas, TAMs represent the most abundant immune cell population [37, 38]. TNF$\alpha$ is part of the repertoire of cytokines secreted by TAMs [38], and therefore we studied macrophages to examine their potential involvement in the regulation of *SULF1* and *HSPG2* expression and to explore their prevalence and phenotype in bone metastases of PCa. In contrast to the binary definition of macrophages as M1 or M2, TAMs represent multiple, distinct subpopulations that share features of both types, but usually behave more like wound-healing and developmental phenotypes, especially in end stages of disease [39–42]. Therefore, we studied macrophages both to explore their prevalence and phenotype in bone metastases of PCa and to examine their potential involvement in the regulation of *SULF1* and *HSPG2* expression in desmoplastic bone marrow stroma. Interestingly, in the patient specimens we tested, intra- and inter-patient heterogeneity among TAMs was observed regarding the expression of phenotypical markers. In four out of five patient samples, we found TAMs expressing CD163, CD206, or both (S6 Fig). Macrophages in triculture with PCa cells C4-2B and fibroblastic HS27A cells only displayed CD206, which suggests polarization towards a specific M2 subpopulation. Also, during tumor progression, *in vivo*, cancer-associated cells interact with the tumor for several years in comparison to our biomimetic model, in which cells interact for days to weeks. Thus, given the high phenotypic plasticity exhibited by macrophages, polarization states are expected to differ depending on signals and cellular interactions in various

contexts. Corroborating our observations, previous studies showed that CD206[+] macrophages are prevalent in prostate tumors in the bone of mice [43, 44].

Here we report that M2-like macrophages can increase transcript levels of *SULF1* and *HSPG2* in bone marrow fibroblasts. Although TAMs and CAFs have been traditionally associated with a pro-tumorigenic behavior [45, 46], these cells react to stimuli *via* complex, context-dependent ways that are not exclusive to potentiating tumor progression [11, 47, 48]. While sequential and coordinated events, such as inflammation, epithelialization, and matrix-remodeling, are required for wound repair, this response is dysregulated and perpetuated in carcinomas [34, 49]. As a result, the stromal and immune reactions to cancers encompass the concomitant activation and suppression of diverse signaling pathways, positively and negatively impacting various aspects of cancer cell behavior at the same time. For example, while the accumulation of perlecan initially functions as a tissue barrier to slow metastatic spread [50], it simultaneously increases the availability of heparan sulfate chains that stabilize factors that can enhance tumorigenicity. Similarly, even though silencing of *SULF1* has been mostly associated with poor prognosis in a variety of carcinomas [33, 51–53], we hypothesize that these outcomes rely heavily on the signaling context, such as type of tumor, disease stage, spatial distribution of GAGases, specific ligands, and other factors [32]. In fact, SULF1 has displayed tumor-promoting activity in pancreatic, urothelial, and gastric cancers [51, 54, 55], which further highlights the complex outcomes of SULF1 activity *in vivo*.

For the first time, we identified the main cellular sources and levels of *HSPG2* and *SULF1* transcripts in the PCa bone metastatic microenvironment and examined the ability of the proteins they encode to regulate Wnt3A-mediated PCa growth in perlecan-modified hydrogels. Previous reports explored *SULF1* expression mostly in cancer cells [56], with the majority showing reduced levels of *SULF1* in progressive ovarian, hepatocellular, breast, and head and neck carcinomas [52, 57–60]. Given that SULFs can act directly to remove 6-O-sulfate from HSPG co-receptors at the cell surface [18, 61], it is reasonable to expect that cancer cells would down-regulate their *SULF1* expression to escape signaling suppression. Consistent with this hypothesis, data gathered in the Prostate Cancer Transcriptome Atlas [62], comprising over 1300 clinical specimens from 38 cohorts, show a significant reduction of *SULF1* expression in metastatic castrate-resistant PCa compared to benign and primary tumors (S11 Fig). Furthermore, SULF1 protein levels were found to be higher in the stromal compartment of primary prostate tumors compared to their healthy counterparts [63], which is consistent with our observations in bone metastatic sites. At first it seemed counterintuitive that SULF1 levels are reduced in the metastatic-castrate resistant prostate cancer (mCRPC) specimens, but continue to be expressed in CAFs, as demonstrated here. Human PCa specimens from The Prostate Cancer Transcriptome Atlas [62] likely contain both tumor and infiltrated stromal cells, however, the number of cancer cells far exceeds the number of stromal cells. Thus, the overall downregulation of SULF1 from benign to advanced forms of PCa is irrespective of stroma-derived SULF1 and is consistent with our findings that loss of SULF1 accelerates PCa cancer growth in bone metastases.

Despite its importance in Wnt3A signaling, the impact of SULF1 activity, either at primary or metastatic sites, had never been investigated prior to this work. Based on our findings here, we present SULF1 as an additional negative regulator of Wnt3A-mediated/induced growth in bone-adapted metastatic PCa cells. In our 3D triculture models, even in the presence of M2-like macrophages and exogenous Wnt3A, the presence of SULF1 was sufficient to prevent Wnt3A-driven growth stimulation of C4-2B cells. This effect is most likely attributable to SULF1-mediated removal of 6-O-sulfate from heparan sulfate chains on cell surface proteoglycans. This activity disrupts ternary signaling complexes in which heparan sulfate acts as a co-factor to stabilize and potentiate signal transduction mediated by receptors with heparan

sulfate proteoglycan co-receptors. Interestingly, Fellgett and colleagues [64], demonstrated that SULF1 can have opposite effects on Wnt/$\beta$-catenin signaling depending on the type of Wnt involved. In contrast to our findings here, they showed that over-expression of both SULF1 and Wnt3A did not affect Wnt3A-mediated activation of the canonical signaling pathway in *Xenopus*, while also demonstrating that SULF1 inhibits Wnt8a- and enhances Wnt11b-mediated signaling. Curiously, Tang and Rosen [28] found that HEK293 cells over-expressing both SULF1 and Wnt3A displayed a 2-3-fold increase in Wnt signaling compared to a group over-expressing Wnt3A only, suggesting that SULF1 promotes signaling in this context. These seemingly contradictory results may largely be explained by differences in experimental settings. To assert SULF1 functions as a tumor suppressor, we considered not only its direct impact in our 3D tricultures but also its expression levels by cell lines and metastatic tissues, in addition to the influence of tumor-associated cells regulating *SULF1* and *HSPG2* expression.

We propose a model that illustrates how stromal-derived SULF1 in the reactive stroma microenvironment inhibits Wnt3A-mediated growth of PCa cells (Fig 6). SULF1 produced and secreted in the stroma can remove key sites of sulfation from heparan sulfate that are needed to stabilize Wnt3a and that are crucial for the co-receptor functions of heparan sulfate at the cell surface, thus limiting signaling. Because loss of *SULF1* increases Wnt3a-induced growth, it suggests that the cell surface retention of 6-O-sulfate is critical for growth stimulation, as also demonstrated by Ling and collaborators [23]. Therefore it seems that loss of *SULF1* increases 6-O-sulfate retention at the cancer cell surface and allows the formation of a greater number of stable ternary complexes for robust signal transduction.

In mCRPC, although *SULF1* is nearly silenced in cancer cells, the macrophage infiltrated tumor stroma continues to produce SULF1 and perlecan. Although we saw a growth suppression role for stromal-derived SULF1 in the context of Wnt3A signaling, it is possible that signal transduction of other cancer-promoting pathways are concomitantly potentiated. SULF1 is not a global signaling inhibitor [64, 65] and any biological impact of its activity has to be considered in view of the signaling context. The general trend, however, indicates that *SULF1* downregulation is associated with worse prognosis in several cancers. In advanced forms of PCa, as seen in metadata from human benign and primary lesions [62], the loss of *SULF1* is suggested to follow the transition from latent to lethal disease. Like the well-documented impact of the loss of tumor suppressors *PTEN* [66] and *TP53* [67] in PCa, we conclude that *SULF1* loss also removes an important brake on tumor growth at later stages of the disease.

## Materials and methods

### Cell Lines and patient-derived xenografts

LNCaP, C4-2, C4-2B (gifts from Dr. Leland Chung, UCLA) and HS27A and PC-3 (ATCC, VA) cells were maintained in RPMI-1640 with 2 mM L-glutamine and 25 mM HEPES (ThermoFisher, MA) supplemented with 10% (v/v) heat-inactivated fetal bovine serum (FBS; Atlanta Biologicals, GA) and 1% (v/v) penicillin-streptomycin solution (P/S; ThermoFisher). HS-5 (ATCC), WPMY-1 (ATCC) and human primary bone marrow stromal cells (BMSC; obtained as formerly described [68]) were cultured in DMEM with 4 mM L-glutamine and 25 mM D-glucose (ThermoFisher) supplemented with 10% (v/v) FBS and 1% (v/v) P/S. Media containing FBS and P/S or no additives will be referred to as "complete" or "base", respectively. The MDA PCa 118b and 183 PDXs (gifts from Dr. Nora Navone, UT MD Anderson Cancer Center) were routinely maintained in complete RPMI, and further information about history and characteristics were described by Zhi and colleagues [69]. All cells used in this work were routinely tested and shown to be negative for mycoplasma using commercial kits (Millipore-Sigma, MP0025-1KT). Cell lines used in this work either were authenticated by ATCC prior to

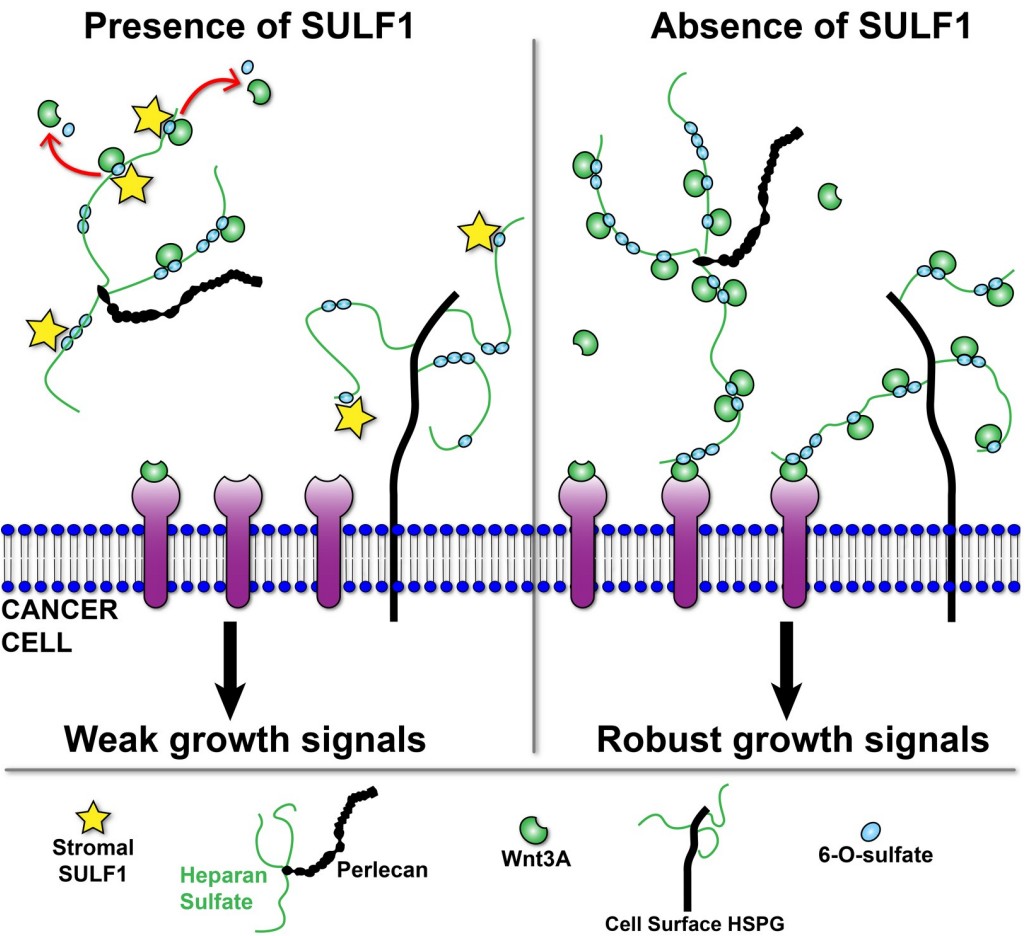

**Fig 6. Model for SULF1 tumor-suppressing role in PCa biomimetic hydrogels.** Stroma-derived SULF1 (yellow stars) promote 6-O-desulfation of heparan sulfate (HS) chains (thin green lines) on HSPGs in the matrix, such as perlecan, and at cell surface. This action results in reduced signaling due to 1) releasing of HBGFs from HSPGs, and 2) disturbing formation of ternary signaling complexes. Lack of SULF1 in the microenvironment allows HBGFs to remain stabilized on HSPGs and to activate receptors at the cell surface, promoting signaling.

expansion in our laboratory, by STR profiling analysis, or by microarray sequencing. Primary BMSCs were identified by expression of the phenotypic marker STRO-1. All cells were fed every 2-3 days. The cell lines were used at passage $\leq$ 15. Primary BMSCs and PDXs were used at passage $\leq$ 5.

## RNA isolation and quantitative reverse-transcriptase PCR

Prostate cancer and immortalized cells were cultured in 6-well plates until 90% confluent; PDX cells were grown in suspension *via* rotation culture (55 rpm) in 6-well plates, $\sim$ 300,000 cells/well, for 48h; and macrophages (seeded at $\sim$ 120,000 cells/cm$^2$ on 6-well plates) were on day 7 of polarization (M1-M$\phi$, M2-M$\phi$) or remained unpolarized (M0-M$\phi$). Total RNA was extracted from cells in triplicate, using the Direct-zol RNA Miniprep kit (Zymo Research, CA; R2052). Briefly, samples in TRIzol™ are mixed 5:1 with chloroform (MilliporeSigma, MA; 472476), incubated at room temperature for 3 minutes, and centrifuged at 12000 g, for 15 minutes, at 4°C. Then, the RNA-containing aqueous phase is mixed 1:1 with 100% ethanol

(MilliporeSigma; 459836), transferred to the miniprep columns, and the rest of the steps are performed per manufacturer's instructions. Genomic DNA was eliminated through DNAse I digestion using the TURBO DNA-*free*™kit (ThermoFisher; AM1907). Reactions were performed in three technical replicates for each biological replicate, using 100 ng of total RNA per 25 $\mu$L reaction with the qScript™One-Step SYBR® Green qRT-PCR kit (QuantaBio, MA; 95057-200). Thermocycling program: cDNA synthesis at 50˚C for 10 min, Taq activation at 95˚C for 5 min, 40 repetitions of 95˚C for 10 seconds and 60˚C for 30 seconds, and lastly a melting curve step. All primers were purchased from Integrated DNA Technologies (IDT; IA). Primer details and sequences can be found in Table 1. Reactions were performed in a CFX96 Real-Time PCR Detection system (Bio-Rad Laboratories Inc., CA). *GAPDH* was used to normalize and calculate the relative amounts of mRNA through the ΔΔCt method (Schmittgen et al. 2008).

## Tissue procurement and processing

Bone marrow tissue specimens containing PCa tumors were obtained under Institutional Review Board (IRB) approved protocols at the Knight Cancer Biolibrary at the Oregon Health and Science University (Portland, OR), the Sepulveda Research Corporation VA Medical Center (Los Angeles, CA), and the University of Michigan, Rapid Autopsy Program (Ann Arbor, MI). The collection of primary bone marrow stromal cells was conducted at the Christiana Care Health System (Newark, DE) under IRB approval. Prior to the collection of the tissues, all institutions received written informed consent from each subject, in accordance with the Declaration of Helsinki. The samples were provided to us fully de-identified, with a material transfer agreement approved by the Committee for the Protection of Human Subjects at The University of Texas. The bone tissue specimens were decalcified using Formical-4™(StatLab, TX, USA), fixed with formalin and paraffin-embedded (FFPE), and provided as sections of 10-15 microns on glass slides. In this study we used samples from five different patients, with age ranging from 50 to 76 years old, collected no earlier than 5 years ago, showing less than 5% necrosis and composed of at least 50% PCa tumor cells. All personally identifiable information was kept confidential by the source institutions.

## Immunostainings

FFPE tissue was deparaffinized and rehydrated in 3 x 5 min changes of xylenes, 2 x 3 min changes each of 100%, 95%, and 70% histology grade ethanol, ending with one 5 min wash with ultrapure water. Heat-induced antigen retrieval was performed by incubating slides in a solution of 10 mM sodium citrate pH 6 (MilliporeSigma; SX0445-20) with 0.05% (v/v) Tween20 (ThermoFisher) at 98˚C for 15 min. Slides were kept in retrieval solution until

**Table 1. Primers used for qRT-PCR experiments.**

| Gene | Amplicon | Forward 5'-3' | Reverse 5'-3' |
|---|---|---|---|
| *SULF1* | 74 bp | AGACCTAAGTCTTGATGTTGGAA | CCATCCCATAACTGTCCTCTG |
| *SULF2* | 194 bp | TGAGGGAAGTCCGAGGTCAC | CTTGCGGAGTTTCTTCTTGC |
| *HPSE* | 108 bp | CTCGAAGAAAGACGGCTAAGA | TGGTAGCAGTCCGTCCATT |
| *HSPG2* | 94 bp | CAATTGTGAGGAGCCAGTC | TGAGAGTGCGTGCTTGCTTTC |
| *CD163* | 138 bp | AGCATGGAAGCGGTCTCTGTGATT | AGCTGACTCATTCCCACGACAAGA |
| *CD206* | 84 bp | CGAGGAAGAGGTTCGGTTCACC | GCAATCCCGGTTCTCATGGC |
| *CD80* | 77 bp | CTGCCTGACCTACTGCTTTG | GGCGTACACTTTCCCTTCTC |
| *GAPDH* | 117 bp | TTGAGGTCAATGAAGGGGTC | GAAGGTGAAGGTCGGAGTCA |

reaching room temperature and washed 2 x 5 min with Tris-buffered saline (TBS; Thermo-Fisher) plus 0.3% (v/v) Triton X-100 (TBSTx). A hydrophobic barrier was drawn around the tissue using a PAP pen. Blocking was carried in TBSTx containing 10% (v/v) goat serum and 1% (w/v) bovine serum albumin (BSA) for at least 60 min. Primary antibodies were diluted in blocking buffer and incubated overnight at 4˚C, at the following dilutions: anti-CD163, 1:400 (Novus Biologicals, CO; NBP1-30148); anti-CD206, 1:400 (BioLegend, CA; 321102); anti-αSMA, 1:200 (Abcam, United Kingdom; ab5694); and E-cadherin, 1:200 (Cell Signaling Technology, MA; 3195S). Then, samples were washed 3 x 5 min with TBSTx, followed by 60 min incubation with secondary antibodies AlexaFluor™ 488 (ThermoFisher; A11029) and 568 (ThermoFisher; A11036), diluted 1:1000, and 4',6-diamidino-2-phenylindole at 5 $\mu$g/mL (DAPI; ThermoFisher) in TBS 1% BSA. Then, slides were washed 3 x 5 min with TBSTx, and coverslips were mounted after adding 1-2 drops of ProLong™ Anti-fade reagent (Thermo-Fisher; P10144). The mountant cured for 24 hours to achieve the best refractive index before imaging. The following steps are for the immunostaining of hydrogels constructs. Hydrogels were rinsed with TBS, fixed with 4% (w/v) paraformaldehyde for 10 minutes, and washed 3 x 5 min with TBS. Cells were permeabilized during a 5 min incubation with TBSTx, followed by blocking with TBSTx containing 10% goat serum for 60 min. Primary antibodies were diluted in blocking buffer and incubated overnight at 4˚C, at the following dilutions: anti-CD163, 1:200 (Novus Biologicals, CO; NBP1-30148); anti-CD206, 1:200 (BioLegend, CA; 321102); anti-Vimentin, 1:300 (Abcam; EPR3776); and anti-E-Cadherin, 1:100 (Cell Signaling Technology, MA; 3195S). Samples then were washed 4 x 5 min with TBSTx followed by incubation with secondary antibodies, AlexaFluor™ 488 and 568 diluted 1:500 in TBS 1% BSA, for 60 min. Hydrogels then were washed 4 x 5 min with TBSTx and counterstained for 10 min with phalloidin probe AlexaFluor™ 647, 1:40, (ThermoFisher; A22287) and DAPI 5 $\mu$g/mL in TBS. Two more washes with TBS were performed and hydrogels were kept immersed in TBS, protected from light until imaging. All steps were performed at room temperature unless otherwise indicated. All antibody incubation and wash steps were performed without or with agitation (50 rpm), respectively. Immunofluorescence images were captured in an automated fashion, with a Nikon A1-Rsi confocal microscope (Nikon Corporation, Japan), using the NIS-Elements JOBS module (Nikon Corporation). For the tissue sections, the entire area of each sample was automatically captured with 20X objective, generating 15-micron Z-stacks, 3-micron steps. For hydrogels, using the 20X objective, four 150-micron Z-stacks, 5-micron steps, at random regions were acquired in each hydrogel.

## RNAscope assay for *in situ* RNA detection

FFPE bone tissue with PCa metastases was subjected to RNA *in situ* hybridization (RISH). The RNAscope® Intro Pack 2.5 HD Reagent Kit Brown-Hs (ACDbio, CA; 322370) was used following the manufacturer's instructions exactly. The HybEZ™II system was used for all incubation steps warmer than room temperature. The hybridization probes were designed and purchased from ACDbio: SULF1-Hs (403581), HSPG2-Hs (573501), Wnt3A-Hs (429431). Negative, DapB, and positive, PPIB, control probes were used in every experiment. After the signal development step (chromogenic, DAB), each section was counterstained for exactly 10 seconds with hematoxylin QS (Vector Laboratories, CA; H-3404), and lastly, mounted in Cytoseal™60 (Thermo; 8310-16). After curing of the mountant, the entire section for each sample was imaged via color brightfield, 40X objective, using the Keyence automated microscope BZ-X8100 (Keyence, Japan).

## RISH–Immunofluorescence multiplexing

RISH–Immunofluorescence multiplexing First, RISH was performed on serial FFPE tissue sections according to the RNAscope manufacturer's instructions, with the following exceptions. The RNAscope® Protease Plus Reagent solution was diluted 1/5 in sterile, Milli-Q water, and the incubation time was reduced to 15 min. After hematoxylin counterstaining, immunofluorescence was conducted as described above, starting with the blocking step and ending with mounting coverslips using ProLong™ Anti-fade reagent. After 24 hours, color brightfield and immunofluorescence images were acquired using a 40X objective in the Keyence automated microscope BZ-X8100 (Keyence).

## Image processing and analysis

ImageJ (FIJI, version 1.52p), developed at the National Institutes of Health (USA) and Imaris (Bitplane, CT) were used for image processing and quantifications. In Fig 2, to process the *SULF1* and *HSPG2* mRNA staining, the chromogenic signal was extracted using color deconvolution and the "H DAB" setting. The generated image then was adjusted by thresholding (Otsu, B&W) until the signal visually matched the original chromogenic staining, but the software was incapable of deconvolving 100% of the signal. The same thresholding settings were applied to all images so that the margin of error was consistent. For each region, the deconvolved signal (red channel) was merged with the green ($\alpha$SMA), gray (E-cad), and blue (DAPI) channels. Imaris was then used for the quantification of the mRNA signals. $\alpha$SMA$^+$ and E-cad$^+$ pixels were assigned surfaces, whereas *SULF1*$^+$ or *HSPG2*$^+$ pixels were assigned spots. Then, with the raw numbers, the percentage of spots in $\alpha$SMA$^+$ versus E-cad$^+$ cells was measured. In Fig 5, each triculture group was stained with E-cad and DAPI or vimentin and DAPI, as described above. The total cellularity of C4-2Bs, in samples stained with E-cad, was quantified by assigning DAPI$^+$ pixels as spots, filtering out all spots in E-cad- cells. To assess C4-2B cellularity in hydrogels stained with vimentin, all DAPI spots were counted, excluding the ones within vimentin$^+$ cells. To measure the volume of each PCa tumoroid, E-cad positive pixels were calculated as surfaces. Clusters with fewer than 90,000 $\mu$m$^3$ were excluded to eliminate single cells and small aggregates (2-5 cells). Additionally, because the cell sphericity of fibroblasts is significantly lower than that of macrophages, a sphericity filter was employed based on vimentin staining to allow pseudo coloring of macrophages (yellow) with high confidence. The sphericity of macrophages stained with CD68, CD163, and CD206 was used as a threshold, and vimentin-positive cells undergoing evident cell division were excluded.

## Isolation of primary human monocytes

De-identified peripheral blood samples (Leukopaks) were obtained from healthy adult male donors at the Gulf Coast Regional Blood Center (Houston, TX, USA). The work was conducted with approval from Rice University's IRB. The anticoagulated blood was obtained at room temperature, no longer than 24 hours after draw, and immediately processed for isolation of peripheral blood mononuclear cells (PBMCs) by density gradient centrifugation using Ficoll-Paque Premium 1.077 g/mL (GE Healthcare, IL; 17544202) according to the manufacturer's instructions. PBMCs were either processed for sorting of monocytes or frozen (5 x 10$^7$ cells/mL) in complete RPMI with 5% (v/v) dimethyl sulfoxide. Monocytes were sorted from PBMCs *via* magnetically-activated cell sorting (MACS) through negative selection, using the Pan Monocyte Isolation Kit (Miltenyi Biotec, Germany; 130-096-537) and autoMACS Separator Pro or MiniMACS separator (Miltenyi Biotec), following manufacturer's instructions.

## Maturation and polarization of macrophages

Immediately after sorting and counting, monocytes were resuspended in base RPMI (37˚C), seeded at 120,000 cells/cm$^2$ and incubated at 37˚C in a humidified atmosphere of air:CO2 (95/5; v/v) for 30 min to allow the monocytes to attach. Monocytes were seeded onto different culture vessels depending on the downstream experiment—for RNA isolation, monocytes were seeded onto 6 well plates; to harvest mature macrophages for hydrogel encapsulation, it was critical to seed monocytes onto non-tissue cultured treated 100 mm plastic dishes (Thermo-Fisher). After the 30 min incubation, the attached monocytes were washed 3x with warm (37˚C) Ca$^{2+}$/Mg$^{2+}$-free PBS (Lonza, Switzerland) and the medium was replaced with complete RPMI supplemented with 50 ng/mL of either macrophage colony-stimulating factor (M-CSF; 574804), to generate unpolarized (M0) and alternatively activated (M2) macrophages, or granulocyte-macrophage colony stimulating factor (GM-CSF; 572902), to obtain classically activated macrophages (M1). M-CSF or GM-CSF were refreshed in feedings on days 2 and 4. M2-polarized cells were obtained by treating cells, on day 4, with 50 ng/mL of interleukins (IL) 4 (574002) and 13 (571102) and 40 ng/mL of IL-10 (571002). M1 polarization was achieved by treatment with 50 ng/mL of interferon-gamma (IFN-$\gamma$; 570202) on day 4, followed by 100 ng/mL of lipopolysaccharides (MilliporeSigma; L4391-1MG), directly added to culture on day 6. All cytokines were purchased from BioLegend, in recombinant human format, unless otherwise stated. Confirmation of macrophage polarization was done by both qRT-PCR, for which we used primers to quantify mRNA levels of *CD163*, *CD206* and *CD80*, and immunofluorescence of CD68, CD163, and CD206 in macrophages within hydrogels (See Immunostaining section for details).

## Preparation of hydrogels for 3D cell culture

To support the growth, interactions, and viability of cells used in this study, we created a custom hydrogel consisting of hyaluronic acid (HA) (Advanced BioMatrix, CA; GS220), 3D culture collagen I (Trevigen, MD; 3447-020-01), and recombinant human perlecan domain I (PlnDmI), prepared as described in [70]. For each Wnt3A containing hydrogel, 200 ng of recombinant human Wnt3A (R&D Systems, MN; 5036WN010) were conjugated to 12 $\mu$g of PlnDmI in 20 mM HEPES, pH 7.6, for 3h at room temperature. The hydrogel solution was prepared as follows. First, 4˚C ultrapure water, 4˚C 10X PBS, and 4˚C 1 M freshly prepared sodium hydroxide (NaOH), were mixed into an Eppendorf tube according to instructions from the collagen type I manufacturer. Then, collagen type I was added, followed by HA and PlnDmI (the volumes of HA and PlnDmI are subtracted from the water volume). Final concentrations for both collagen and HA were 1 mg/mL, and 60 $\mu$g/mL for PlnDmI. The solution was kept on ice (no longer than 1 hour) until resuspension with cells. If necessary, more NaOH was added to adjust the pH to 7.6 (this is crucial for gelation). The final volume of hydrogel solution varied per experiment depending on the number of hydrogels needed. Cells were detached from their culture flasks by incubation with trypsin-EDTA 0.25% (Thermo-Fisher) at 37˚C for 5 min or, for macrophage detachment, by incubation with PBS supplemented with 5 mM EDTA (ThermoFisher) 30 min at 4˚C. Then, cells were counted and resuspended in 200 $\mu$L of hydrogel solution. In triculture experiments, seeding densities per hydrogel were as follows: 40,000 cancer cells, 10.000 fibroblasts, and 5.000 for macrophages. The hydrogel-cell mixtures were incubated in vials at 37˚C for 2-3 min (this kickstarts gelation and prevents cell settling), resuspended again, and cast into wells of a 48-well plate or 35-mm dishes with #1.5 thick coverslip bottom (Cellvis, CA; D35141.5N). The continuation of gelation took place under incubation at 37˚C in a humidified atmosphere of air:CO2 (95:5, v/v) for at least 30 min. Then, 500 $\mu$L or 1.5 mL of complete RPMI (at 37˚C) were added to each well or

35-mm dishes, respectively. The hydrogel samples were placed back in the incubator, and cells were fed every 2-3 days. After encapsulation, viability was assessed on days 1, 3, and 7 using CalceinAM (Biotium, CA; 148504-34-1), Ethidium Homodimer I (Biotium; 61926-22-5) per manufacturer protocol and expression of phenotypical markers was measured at day 5 *via* immunostaining. Hydrogels prepared for the experiments in Fig 5 were cultured in RPMI with 2% (v/v) FBS and 1% (v/v) P/S.

## Conditioned medium experiments

To obtain conditioned medium of M2-M$\phi$, monocytes were seeded and cultured as described above. On day 7 of culture, the culture medium was discarded, cells were washed twice with PBS, and cultured for another 48h in complete RPMI. Then, the conditioned medium was collected and centrifuged at 1000 g for 10 minutes and sterile filtered. Primary BMSCs, HS27A, and C4-2B cells were seeded in triplicate onto 6-well plates so that they would be approximately 70% confluent on the day of collection of M2-M$\phi$ conditioned medium, allowing the fresh use for every experiment. Cells were exposed to a mixture of 50% fresh complete RPMI and 50% M2-M$\phi$ conditioned medium for 48h, after which RNA was isolated and processed as described in the RNA isolation section.

## CRISPR/Cas knockout of *SULF1* in HS27A cells

The Gene Knockout Kit for was purchased from Synthego (CA, USA), which included single guide RNAs (sgRNAs) for *SULF1*, Cas9 enzyme (*Streptococcus pyogenes*) and a validated sgRNA for RELA to serve as a positive control. The kit was used to deliver the sgRNA-Cas9 ribonucleoproteins (RNPs) to HS27A cells with Lipofectamine CRISPRMAX reagent (ThermoFisher; L3000001), according to the protocol provided by Synthego. Reverse transfection was the method of choice for delivery of Lipofectamine-RNP complexes, which were added to wells of 12-well plates, followed by the addition of HS27A in suspension, at 80-100,000 cells/ well or 20-25,000 cells/cm$^2$. mRNA was isolated 48h post-transfection, and qPCR was performed for gene expression analysis. Also at the 48h time-point, other replicate groups were used for isolation and expansion of multiple monoclonal populations *via* limiting dilution, until we obtained the *SULF1* knockout HS27A line. Potential clones were screened through qPCR, in addition to the tool Inference of CRISPR Edits (ICE) developed by Synthego (https:// ice.synthego.com) (S12 Fig).

## Western blot

Protein was extracted from cells using RIPA Lysis Buffer (ThermoFisher) and Halt Protease Inhibitor Cocktail (ThermoFisher), per manufacturer instructions, then mixed with 4X NuPAGE™ LDS Sample Buffer (ThermoFisher) and subsequently incubated at 70˚C for 10 min. Electrophoresis was performed using NuPAGE™ 10% (w/v) polyacrylamide Bis-Tris gels (ThermoFisher), loaded with 20-30 μg of total protein per lane. Gels were run using 1X NuPAGE™ MOPS SDS Running Buffer (ThermoFisher) at 180 V for 50 min. Transfer was done via the wet tank method, using 1X NuPAGE Transfer Buffer (ThermoFisher) with 10% (v/v) or 20% (v/v) methanol (depending on whether 1 or 2 gels were used, respectively) and transferred into 0.45 μm pore size nitrocellulose membranes (GE Healthcare) at 4˚C, 40 V for 5 hours, using a magnetic stirrer at 300 rpm. Membranes then were washed in ultrapure water to remove Ponceau and immediately blocked with TBS 0.01% (v/v) Tween-20 (TBST) with 5% (w/v) non-fat dry milk (Bio-Rad Laboratories) for at least 1 hour with gentle rocking at room temperature. The membranes then were incubated overnight with primary antibodies in blocking buffer, with gentle rocking at 4˚C. Subsequently, the blots were washed 4 x 5 minutes

with TBST, rocking, followed by incubation with secondary antibodies (Fluorescent* or HRP-conjugated) in blocking buffer for 1-1.5 hours at room temperature with gentle rocking. After 4 x 5 min washes with TBST and 1 x 5 min wash with TBS, blots were developed either with chemiluminescence or detection of near-infrared fluorescence, which are indicated in the figures. For chemiluminescence, we incubated blots with the SuperSignal™ West Dura ECL (ThermoFisher) substrate as described by the manufacturer, and exposed to HyBlot CL autoradiographic films (Denville Scientific Inc., MA, USA). For near-infrared fluorescence detection, after the final wash, membranes were scanned with the Odyssey Classic Imager (LI-COR). The following secondary antibodies were used. For NearIR detection, donkey anti-mouse IRDye 680LT (LI-COR, NE, USA, 925-68020) or IRDye 800CW (LI-COR, 925-32211) were used at 1:20000. For chemiluminescence detection, HRP-conjugated sheep anti-mouse (Jackson Laboratory, ME, USA) or goat anti-rabbit were used at 1:50000 (Abcam, ab97051). *Protected from light.

## Statistical analysis

All bar graphs represent means ±SD of at least triplicate samples and are representative of at least two independent experiments. Quantitative PCR was analyzed using unpaired, two-tailed Student's T-test. The quantification of PCa cellularity and cluster formation in 3D hydrogels was analyzed by one-way ANOVA with Tukey post-test. Differences were considered significant at $p < 0.05$. Statistical analyses were performed with GraphPad InStat (GraphPad Software, CA).

## Supporting information

**S1 Fig. Basal mRNA expression of *SULF1*, *SULF2* and *HPSE* in bone marrow stromal fibroblasts.** Cells at about 90% confluency were lysed with TRIzol and RNA was isolated for qRT-PCR as described in Materials and Methods. Displaying the data by comparing between SULFs and HPSE indicates more clearly that *SULF1* is the major heparan sulfate modifying enzyme expressed by either the bone marrow fibroblast cell line HS27A (A), or primary cultures of bone marrow fibroblasts (BMSC) (B). The expression of *SULF1*, *SULF2* and *HPSE* was normalized to that of *GAPDH*. Values obtained for *HPSE* were arbitrarily set to 1 for comparison. Data shown represent the mean ±SD of two independent experiments.
(TIF)

**S2 Fig. Testing of multiple SULF1 antibodies.** In blots from A-F: 1—SULF1-KO-HS27A cells, 2—WT-HS27A cells, and 3—C4-2B cells. Western blot was conducted as described in Materials and Methods. Approximately 20 $\mu$g of total protein lysate was loaded per well. All the antibodies tested are listed under each blot. All antibodies show one or multiple bands, all of which are inconsistent with the predicted molecular weight of 100-125 kDa for SULF1. The amounts used were equal to the highest concentrated dilution recommended by the manufacturers. Blots A-C were developed via enhanced chemiluminescence and films were exposed for approximately one minute. Blots D-F were developed by fluorescence detection using secondary antibodies labeled with near infra-red flourophores, as described in Materials and Methods. The raw, uncropped images can be found in file "S1 Raw images".
(TIF)

**S3 Fig. RNA *in situ* hybridization (RISH) and immunofluorescence multiplexing in serial sections of cervical spine with PCa metastases.** (A-D) RISH–immunofluorescence multiplexing of *SULF1* (A1-6 and C1-6) and *HSPG2* (B1-6 and D1-6) with the stromal marker $\alpha$SMA (green) and epithelial marker E-cad (gray). A and B were magnified for display in Fig 2 and

represent distinct regions of serial sections probed with the respective markers. C and D represent the same region of serial sections probed with the respective markers. The chromogenic signal in panels A1, B1, C1, and D1 was deconvoluted in ImageJ to create images A2, B2, C2, and D2, as described in Materials and Methods. RISH–immunofluorescence multiplexing was performed as described in Materials and Methods. Nuclei are stained with DAPI (blue). Scale bars correspond to 40 $\mu$m. Images were acquired using 40x objectives. Four additional images were acquired per sample, at random regions, which were used for quantification of *SULF1* and *HSPG2* signals, as described in Materials and Methods.
(TIF)

**S4 Fig. RNA *in situ* hybridization staining from high and low magnification areas of bone with PCa tumors.** This figure shows the mRNA expression of the positive control gene, (A) *Peptidyl-prolyl cis-trans isomerase B* (*PPIB*), (B) *SULF1*, and (C) *HSPG2* in the same region of cervical spine specimens. In D, E, and F, femur samples also are probed and include the negative control gene *dihydrodipicolinate reductase* (*dapB*). *PPIB* expression was widespread in all cells of the tissue, indicating good quality of the mRNA in the sample, whereas *SULF1* and *HSGP2* were generally confined to the stroma surrounding tumor nests. The PPIB control was used on every independent replicate experiment. The RNAscope assay was performed as described in Materials and Methods. Scale bar represents 200 $\mu$m for A-C and 50 $\mu$m for D-F.
(TIF)

**S5 Fig. RNA *in situ* hybridization of *SULF1* and *PPIB* in bone marrow of additional patients.** In situ hybridization for *PPIB* mRNA (A1-4) was performed in all hybridization experiments as a positive control for the assay. Samples which failed to show PPIB mRNA signal were disregarded for further analyses. As demonstrated above, *SULF1* signal (B1-4) is mostly confined to the reactive bone marrow stromal cells, while *PPIB* is expressed throughout the serial sections. The tissues of origin for the samples used were femur (A1 and B1), cervical spine (A2, B2, A3, and B3) and acetabulum (A4 and B4).
(TIF)

**S6 Fig. Immunostaining of $\alpha$SMA, CD163 and CD206 in other patients.** Immunofluorescence staining is shown for PCa bone metastasis samples from additional patients. As shown above, reactive bone marrow fibroblasts show strong $\alpha$SMA signal (A1-4). The macrophage infiltration, indicated by the CD163 and/or CD206 staining (B1-4), varied depending on the patient, but the phenotype was generally consistent with polarization towards tumor-promotion (M2-like) macrophages. Larger yellow boxes indicate insets amplified from the smaller yellow boxes within the same figure. The tissues of origin for the samples used were femur (A1 and B1), cervical spine (A2, B2, A3, and B3) and acetabulum (A4 and B4).
(TIF)

**S7 Fig. *CD80* and *CD206* mRNA levels in macrophages cocultured indirectly with C4-2B and HS27A cells.** A. An indirect coculture system was designed in which a PDMS (grey) mold with laser-cut wells were placed in 100-mm dishes. The area of each well was 9 mm$^2$ and the thickness of the mold was 3 mm. Culture combinations were as illustrated. As described in Materials and Methods, RNA was collected from unpolarized macrophages (M$\phi$-Ctrl), and M1- or M2-polarized macrophages (M1-M$\phi$ and M2-M$\phi$, respectively). B. Signals produced by C4-2B alone or C4-2B and HS27A cells did not drive macrophages towards the classically activated phenotype, indicated by CD80 mRNA expression. C. In contrast, while C4-2B cells alone could not induce up-regulation of *CD206*, factors produced by C4-2B and HS-27A cells up-regulated expression of *CD206*, indicating polarization towards a TAM-like phenotype. RNA isolation and macrophage polarization were performed as described in the materials and

methods section. The expression of *CD80* (B) and *CD206* (C) was normalized to that of *GAPDH*. Values obtained for the Mφ Ctrl group were arbitrarily set to 1 for comparison. Data shown represent the mean ±SD of two independent experiments. **, P < 0.01.
(TIF)

**S8 Fig. *SULF1*, *SULF2* and *HPSE* expression in C4-2B cells treated with CM from M2-polarized macrophages.** Human primary monocytes were polarized to M2-Mφs for 7 days and CM was collected on day 9, as described in Materials and Methods. C4-2B cells were treated with CM for 48h. RNA then was isolated for real-time qPCR. The treatment of C4-2B cells with macrophage conditioned medium did not lead to any significant changes in the expression of *SULF1*, *SULF2* and *HPSE*. The expression *SULF1*, *SULF2* and *HPSE* mRNA was normalized to that of *GAPDH*. Values obtained for untreated cells (-) were arbitrarily set to 1 for comparison. Data shown represent the mean ±SD of two independent experiments.
(TIF)

**S9 Fig. *SULF2* and *HPSE* transcript levels in wild-type (WT) and *SULF1*-KO-HS27A cells.** RNA was isolated from WT-HS27A and *SULF1*-KO-HS27A cells at 90% confluence. The knockout of *SULF1* in HS27A cells did not cause significant changes in the expression of *SULF2* or *HPSE mRNA*. The expression *SULF2* and *HPSE* mRNA was normalized to that of *GAPDH*. Values obtained for WT-HS27A cells were arbitrarily set to 1 for comparison. Data shown represent the mean ±SD of two independent experiments.
(TIF)

**S10 Fig. Immunofluorescence of CD206 for detection of macrophages in 3D tricultures.** Biomimetic collagen-HA hydrogels were prepared as detailed in Materials and Methods, identically to the experiment in Fig 5. Immunofluorescence stainings show CD206 (green) with DAPI (blue) counterstain. With the CD206 signal, we created the parameters for the sphericity filter used in Fig 5A2, 5A4, 5A6 and 5A8, to pseudo-color macrophages in yellow. Scalebar = 100 microns. Data shown represent the mean ±SD of three independent experiments. In each experiment, n = 5 hydrogels for each of the four groups.
(TIF)

**S11 Fig. Association analysis of *SULF1* expression with disease course gathered from the Prostate Cancer Transcriptome Atlas.** Expression data can be visualized via box plot (A) or lineplot of mean trend (B), which categorize the patient sample data from benign, local disease to increasing values for the Gleason Score (GS) and mCRPC. These data are consistent with reduction of *SULF1* expression in the most advanced disease stage.
(TIF)

**S12 Fig. Verification of CRISPR-Cas-mediated *SULF1*-knockout.** A. After isolation of multiple clones, we used qPCR to screen for *SULF1* mRNA expression, as described in Materials and Methods. The most complete *SULF1*-KO-HS27A monoclonal population is described here compared to wild-type (WT) HS27A cells. B. DNA was extracted, from both WT and *SULF1*-KO-HS27A cells and sequenced around the CRISPR cut sites (represented by black vertical dotted lines). Sequencing primers were provided in the Gene Knockout Kit by Synthego. The contributions (%) show the inferred sequences present in the *SULF1*-KO-HS27A population. Indel % represents the percentage of sequences with mutations. Our results reveal only two sequences, showing +1 base pair (bp) and -5 bp indels. The Knockout-Score indicates the proportion of cells that have either an indel that causes a frameshift or 21 + bp indel. The score of 100 indicates a complete functional KO of *SULF1* from HS27A cells.
(TIF)

**S1 Raw images.**
(PDF)

## Acknowledgments

We are grateful for the past and current members of the Carson and Farach-Carson laboratories, and for our P01 collaborators, all of whom assisted with numerous helpful discussions during the course of this work. We especially thank Dr. Nora Navone, for providing invaluable prostate cancer PDX cells, Dr. Kenneth Pienta, for presenting us with rare human bone marrow tissue with PCa metastases, and Dr Charles Schneider, for the human primary bone marrow stromal cells.

## Author Contributions

**Conceptualization:** Fabio Henrique Brasil da Costa, Daniel D. Carson, Mary C. Farach-Carson.

**Data curation:** Fabio Henrique Brasil da Costa, Anna Truong, Mary C. Farach-Carson.

**Formal analysis:** Fabio Henrique Brasil da Costa, Michael S. Lewis, Anna Truong, Daniel D. Carson, Mary C. Farach-Carson.

**Funding acquisition:** Fabio Henrique Brasil da Costa, Daniel D. Carson, Mary C. Farach-Carson.

**Investigation:** Fabio Henrique Brasil da Costa, Anna Truong, Daniel D. Carson, Mary C. Farach-Carson.

**Methodology:** Fabio Henrique Brasil da Costa, Anna Truong.

**Project administration:** Fabio Henrique Brasil da Costa, Daniel D. Carson, Mary C. Farach-Carson.

**Resources:** Fabio Henrique Brasil da Costa, Michael S. Lewis, Daniel D. Carson, Mary C. Farach-Carson.

**Supervision:** Daniel D. Carson, Mary C. Farach-Carson.

**Validation:** Fabio Henrique Brasil da Costa.

**Visualization:** Fabio Henrique Brasil da Costa.

**Writing – original draft:** Fabio Henrique Brasil da Costa, Daniel D. Carson, Mary C. Farach-Carson.

**Writing – review & editing:** Fabio Henrique Brasil da Costa, Michael S. Lewis, Anna Truong, Daniel D. Carson, Mary C. Farach-Carson.

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
