## [Decision Letter · Decision Letter 0]

1 Apr 2020

PONE-D-20-05598

SULF1 suppresses Wnt3A-driven growth of bone metastatic prostate cancer in perlecan-modified 3D cancer-stroma-macrophage triculture models

PLOS ONE

Dear Dr. Farach-Carson,

Thank you for submitting your manuscript to PLOS ONE. After careful consideration, we feel that it has merit but does not fully meet PLOS ONE’s publication criteria as it currently stands. Therefore, we invite you to submit a revised version of the manuscript that addresses the points raised during the review process.

Please address the concerns raised by the reviewer.

We would appreciate receiving your revised manuscript by May 15, 2020. To enhance the reproducibility of your results, we recommend that if applicable you deposit your laboratory protocols in protocols.io, where a protocol can be assigned its own identifier (DOI) such that it can be cited independently in the future. For instructions see: http://journals.plos.org/plosone/s/submission-guidelines#loc-laboratory-protocols

We look forward to receiving your revised manuscript.

Kind regards,

Daotai Nie, Ph.D.

Academic Editor

PLOS ONE

Journal Requirements:

2. Please provide additional information about the MDApcA 118B AND 183 PDX cells used in this work, including history, culture conditions and any quality control testing procedures (authentication, characterisation, and mycoplasma testing).

For all other cell lines, please provide additional information about any quality control testing procedures (authentication, characterisation, and mycoplasma testing). For more information, please see http://journals.plos.org/plosone/s/submission-guidelines#loc-cell-lines.

3. Bone marrow tissue specimens containing PCa tumors were obtained with consent under IRB approved protocols at The University of Texas Health Science Center School of Dentistry.'

(a) Please amend your current ethics statement to include the full name of the ethics committee/institutional review board(s) that approved your specific study.  

(b) Once you have amended this/these statement(s) in the Methods section of the manuscript, please add the same text to the “Ethics Statement” field of the submission form (via “Edit Submission”).

4. In the ethics statement in the manuscript and in the online submission form, please provide additional information about the tissue samples used in your study. Specifically, please ensure that you have discussed whether all data and tissue samples  were fully anonymized before you accessed them. If patients provided written informed consent to have data from their medical records used in research, please specifically include this information.

5. Please ensure that you refer to Figure 6 in your text as, if accepted, production will need this reference to link the reader to the figure.

Reviewers' comments:

Reviewer's Responses to Questions

**Comments to the Author**

1. Is the manuscript technically sound, and do the data support the conclusions?

Reviewer #1: Partly

2. Has the statistical analysis been performed appropriately and rigorously? 

Reviewer #1: Yes

3. Have the authors made all data underlying the findings in their manuscript fully available?

Reviewer #1: Yes

4. Is the manuscript presented in an intelligible fashion and written in standard English?

Reviewer #1: Yes

5. Review Comments to the Author

Reviewer #1: Authors of the manuscript examined the role of SULF1 in reactive bone marrow stroma within metastatic bone microenvironment using biomimetic hydrogel incorporating tumor, fibroblast and macrophage cell types. The function of SULF1 in Wnt3A-mediated prostate tumor growth was also examined. Authors concluded that SULF1 loss favors prostate cancer progression by inability to suppress Wnt3A-mediated growth signals in the stroma. Despite of an interesting model with potential relevance to bone metastasis field, there are several issues with this manuscript that need to be addressed by the authors:

1) The significance of studying SULF1 in a context of Wnt3A is not clear. Aside from stating that Wnt3A is a key HBGF implicated in PCa progression there is little rationale provided for investigating the SULF1-Wnt3A axis. Better background needs to be provided by the authors

2) Fig.1 – it is concerning that authors were not able to detect SULF1 protein with any of the available SULF1 antibodies

3) Fig.2 – RNA signal is very difficult to visualize; mock controls need to be shown for RISH

4) Fig.3 – the presence of macrophages in bone marrow tissue is not surprising and has been shown before; the statement that macrophages accumulate around tumors is too strong based on data provided; lack of CD163 expression in the biomimetic model is concerning. CD163 is an M2 marker known to be expressed by the macrophages in the metastatic microenvironment

5) How was experimental concentration of Wnt3A determined for the studies in Fig. 5?

6) If macrophage-derived TNFa is suspected to be a key-contributor to SULF1 expression, this needs to be shown. Do TNFa levels in macrophages change in response to interaction with tumor cells or in triculture?

7) Fig 5- why was not staining for macrophage markers performed rather than using sphericity filter? This is not an accurate way of identifying macrophages, and certainly an approach that does not allow to look at specific phenotypes of macrophages. Not all macrophages are vimentin-positive. What was the significance of only looking at vimentin-positive macrophages?

8) Overall contribution of macrophages to tumor cell growth in this model is overstated. The data, as provided, do not convincingly support the conclusions

9) The model is somewhat confusing. An increase of SULF1 expression in the stroma and loss in the tumor cells is an interesting concept that needs to be investigated. Since the analyses of human databases indicate that SULF1 is lost in metastatic tumors (which one would presume contain both tumor cells and the stroma) – how do the data presented here fil with these findings. This disconnect between SULF1 levels in the stroma vs the tumor needs to be better demonstrated by the authors. The experimental evidence provided here is not very strong.

Minor points:

1) The sentence in Discussion (page 7, line 192) is incomplete: “Here we report that the up-regulation of SULF1 and HSPG2 transcripts in CAFs by M2-like macrophages.”

2) The references to Supplementary Figures in text are not in agreement with Figures and Figure Legends provided. For example: Fig S5 referenced in text should actually be S6, Fig S6 should be S7, and so forth….. In addition, no reference at all is made to Fig S10 in text.

6. PLOS authors have the option to publish the peer review history of their article (what does this mean?). If published, this will include your full peer review and any attached files.

Reviewer #1: No

---

## [Author Response · Author response to Decision Letter 0]

23 Apr 2020

As directed in the e-mail from the editor, Dr. Daotai Nie, we have uploaded here a rebuttal letter (Response to Reviewers.pdf) that responds to each point raised by the academic editor and reviewer.

---

## [Decision Letter · Decision Letter 1]

27 Apr 2020

SULF1 suppresses Wnt3A-driven growth of bone metastatic prostate cancer in perlecan-modified 3D cancer-stroma-macrophage triculture models

PONE-D-20-05598R1

Dear Dr. Farach-Carson,

We are pleased to inform you that your manuscript has been judged scientifically suitable for publication and will be formally accepted for publication once it complies with all outstanding technical requirements.

With kind regards,

Daotai Nie, Ph.D.

Academic Editor

PLOS ONE

Additional Editor Comments (optional):

Reviewers' comments:

Reviewer's Responses to Questions

**Comments to the Author**

1. If the authors have adequately addressed your comments raised in a previous round of review and you feel that this manuscript is now acceptable for publication, you may indicate that here to bypass the “Comments to the Author” section, enter your conflict of interest statement in the “Confidential to Editor” section, and submit your "Accept" recommendation.

Reviewer #1: All comments have been addressed

2. Is the manuscript technically sound, and do the data support the conclusions?

Reviewer #1: Yes

3. Has the statistical analysis been performed appropriately and rigorously? 

Reviewer #1: Yes

4. Have the authors made all data underlying the findings in their manuscript fully available?

Reviewer #1: Yes

5. Is the manuscript presented in an intelligible fashion and written in standard English?

Reviewer #1: (No Response)

6. Review Comments to the Author

Reviewer #1: This reviewer has no additional concerns. Authors adequately responded to the comments and provided additional data to support their conclusions

7. PLOS authors have the option to publish the peer review history of their article (what does this mean?). If published, this will include your full peer review and any attached files.

Reviewer #1: No

---

## [Editor Report · Acceptance letter]

1 May 2020

PONE-D-20-05598R1 

SULF1 suppresses Wnt3A-driven growth of bone metastatic prostate cancer in perlecan-modified 3D cancer-stroma-macrophage triculture models 

Dear Dr. Farach-Carson:

I am pleased to inform you that your manuscript has been deemed suitable for publication in PLOS ONE. Congratulations! Your manuscript is now with our production department. 

With kind regards,

on behalf of

Dr. Daotai Nie 

Academic Editor

PLOS ONE